# Mass Sperm Motility Is Correlated to Sperm Motility as Measured by Computer-Aided Sperm Analysis (CASA) Technology in Farmed Ostriches

**DOI:** 10.3390/ani12091104

**Published:** 2022-04-25

**Authors:** Pfunzo T. Muvhali, Maud Bonato, Irek A. Malecki, Schalk W. P. Cloete

**Affiliations:** 1Department of Animal Sciences, University of Stellenbosch, Private Bag X1, Matieland 7602, South Africa; mbonato@sun.ac.za (M.B.); imalecki@iinet.net.au (I.A.M.); schalkc2@sun.ac.za (S.W.P.C.); 2Directorate Animal Sciences, Western Cape Department of Agriculture Elsenburg, Private Bag X1, Elsenburg 7607, South Africa; 3School of Agriculture and Environment, Faculty of Science, The University of Western Australia, 35 Stirling Highway, Crawley 6009, Australia

**Keywords:** sperm motility, semen assessment, CASA technology, *Struthio camelus*, sperm quality

## Abstract

**Simple Summary:**

Semen quality in ostriches is still being studied to optimize the reproductive performance of these birds in a commercial farming environment. Various methods such as mass sperm motility scoring and computer-aided sperm analysis (CASA) technology have been employed successfully to evaluate semen quality characteristics such as sperm motility. The relationship between these methods in assessing sperm motility in ostriches is unknown. This study evaluated the relationship between the CASA and the mass sperm motility scoring method when analyzing ostrich sperm motility. Positive significant correlations between these methods for sperm motility traits were recorded, suggesting that farmers could utilize the affordable mass sperm motility scoring to screen ostrich males for sperm quality.

**Abstract:**

Semen analyses have gained momentum in various livestock industries. However, in farmed ostriches, semen analysis is still in its experimental stage, and males are not screened for sperm quality before breeding. This study investigated the correlations between computer-aided sperm analysis (CASA) technology and the traditional, yet affordable, mass sperm motility score. Semen was collected from nine South African Black ostrich males (mean age ± SD: 5.25 ± 1.21 years), using the dummy female method for 5 consecutive days monthly, for 8 months. Mass sperm motility scores were recorded on a scale of 1–5 (1: little to no sperm movement; 5: rapid sperm movement). The CASA traits recorded were: total motility (MOT), progressive motility (PMOT), curve–linear velocity (VCL), straight-line velocity (VSL), average path velocity (VAP), amplitude of lateral head displacement (ALH), linearity (LIN), straightness (STR), wobble (WOB), and beat-cross frequency (BCF). The results revealed positive correlations between mass sperm motility and PMOT, MOT, VCL, and VAP ranging from 0.34 to 0.59 (*p* < 0.0001). In contrast, negative correlations were recorded between mass sperm motility and LIN, STR, and BCF, with correlations ranging from −0.20 to −0.39 (*p* < 0.0001). VSL, ALH, and WOB were not correlated to mass sperm motility (*p* > 0.05). Ostrich farmers may thus be able to evaluate sperm motility reliably and potentially select breeding males by using the affordable mass sperm motility scoring method. Determining the correlation between these methods and fertility after artificial insemination or natural mating is however needed.

## 1. Introduction

South Africa contributes up to 70% of total ostrich products worldwide. This makes the South African ostrich industry a leader in ostrich production worldwide. The ostrich industry generates income from the sales of major products such as meat, leather, and feathers. According to Brand and Jordaan [1], it was estimated that out of a total of 420,000 ostriches slaughtered worldwide in the years 2004/05, a proportion of up to 69% originated from the South African industry. However, this industry encounters setbacks such as poor fertility of eggs, poor hatchability, and poor chick survival [2]. Attempts to improve the productivity and effectiveness of the ostrich industry have been initiated through research with assisted reproductive technology, such as artificial insemination, suggested as one of the fundamental elements to address setbacks [3,4]. This technology requires semen to be collected and evaluated for quality before being used. Semen evaluation is of paramount importance in determining male fertility and, thus, reproductive potential [5]. In commercially farmed ostriches, semen evaluation is still in its analytical stages and does not currently form part of the selection for breeder males [6]. One sperm quality trait of interest is sperm motility, which is a measure of sperm functionality and movement ability. The ability of sperm to move through the female reproductive duct is a fundamental aspect to achieve fertility, as only fully functional sperm have the capability to reach the sperm storage site and subsequently the area of fertilization.

Computer-aided sperm analysis (CASA) technology and mass sperm motility scoring are the two main methods currently used in a wide range of livestock species for the assessment of sperm motility. CASA uses quantitative records of individual sperm cell motility [7,8,9,10]. However, this technology presents various challenges to resource-limited farmers: the instrument and software for this technology are costly, require a certain expertise, and need to be validated for quality assurances [11,12]. Furthermore, because of species variation in sperm characteristics, the CASA technology requires species-specific calibration of the measurements linked to an accurate preparation of the samples prior to analysis and validation based on female fertility [11,13]. Mass sperm motility therefore represents an alternative, user-friendly, and affordable method to assess sperm motility. This method only requires a phase-contrast microscope objective with which mass sperm motility is scored based on the vigor of sperm movement [13]. However, this method is regarded as subjective, with results often varying between and within laboratories, as well as being dependent on the assessor’s experience. Interestingly, mass sperm motility was correlated to reproductive efficiency in sheep [13] and poultry [7,14]. Furthermore, sperm motility was highly correlated to fertility in poultry when determined by using the spectrophotometric technique [15]. More recently, sperm motility traits measured by CASA such as straight-line velocity (VSL), average path velocity (VAP), beat-cross frequency (BCF,) and progressive sperm motility (PMOT) of the Japanese quail was correlated to egg fertility traits such as sperm trapped in the outer perivitelline membrane of the egg blastodisc [9]. These studies revealed the usefulness of both methods for the assessment of sperm quality to aid in selecting quality ejaculates for artificial insemination or for selecting specific males for breeding.

In ostriches, sperm analysis recently started after reliable stress-free methods of semen collection were developed [16]. Remarkably, several studies have applied either the mass sperm motility scoring [16,17,18,19] or the CASA method [20,21,22,23,24] to investigate sperm motility in ostriches. These studies reported the effect of the month of semen collection on semen characteristics, suggesting that spring and the summer months resulted in higher quality ejaculates than the winter months [19,21,24]. Specifically, the largest semen volumes and sperm outputs were collected in October and November in South Africa [19]. Sperm motility measured using the subjective scoring method in ostriches was reported to be constant across years [19]. In contrast, sperm motility and sperm kinematic traits recorded using the CASA technology were higher in the summer months than in the winter months [21]. Various studies have also reported between-bird variation and repeatability estimates for semen characteristics [25,26]. However, the repeatability findings were based on small sample sizes and limited to sperm motility measured using the CASA method and not the traditional scoring method. Against this background, there is currently no information on the relationship between these two methods, as well as on the repeatability of sperm motility measures as determined using the traditional scoring method. Should these methods be correlated and repeatable, ostrich farmers could benefit from adopting the mass sperm motility scoring method as the cheaper and quicker method to evaluate breeder males for high sperm motility before using them for breeding purposes.

Therefore, the aim of this study was to investigate the relationship between the mass sperm motility scores and the CASA traits. In addition, the repeatability of the variables’ values assessed by both methods was also determined.

## 2. Materials and Methods

### 2.1. Study Animals

This study was conducted at the Oudtshoorn Research Farm of the Western Cape Department of Agriculture, situated outside Oudtshoorn, South Africa (33°63′ S, 22°25′ E). A total of 9 South African Black (SAB) ostrich males with a mean (±SD) age of 5.25 ± 1.21 years, kept in individual paddocks, were used. These males were trained for semen collection using the dummy female method [16]. The birds were pre-selected for inclusion in this study based on their behavioral responses towards humans [27], breeding values for egg and chick production, as well as their cooperation during semen collection. The dummy female method for semen collection requires male ostriches to direct their sexual behavior towards humans. While the male is at display, the dummy female (a wooden structure covered with a sack and contained an opening where the artificial cloaca is fitted) is pushed gently between the legs of the male to mount and ejaculate inside the fitted artificial cloaca [16]. Ethical approval for this study was granted by the Western Cape Department of Agriculture’s Departmental Ethical Committee for Research on Animals (Ref No.: R9/24).

### 2.2. Semen Collection and Measurements

During an 8-month period in 2020 (February, March, July, August, September, October, November, and December), semen was collected once from each male for 5 consecutive days starting from February until December. Semen volume was measured using an automatic pipette, while sperm concentration was determined using a spectrophotometer (Spectrawave, WPA, S800, Biochrom Ltd., Cambridge, UK) from an aliquot of 20 μL of semen diluted 1:400 (*v*/*v*) with a phosphate-buffered saline solution which contained 10% formalin. The total number of sperm in each semen sample was determined as the product of semen volume and sperm concentration.

### 2.3. Sperm Motility Assessment

Objective sperm motility traits were recorded using the Sperm Class Analyzer® (SCA) version 5.3 (Microptic S.L., Barcelona, Spain) with a Basler A312fc digital camera (Basler AG, Ahrensburg, Germany), mounted on an Olympus BX41microscope (Olympus Optical Co., Tokyo, Japan). A volume of neat semen containing 20 × 10^6^ sperm cells was pipetted into a tube filled with 245 μL of standard sperm motility buffer consisting of sodium chloride (150 mM), TES (20 mM), and 2% male-specific seminal plasma [21]. The tube was then placed in an aerated incubator (BL°CKICE Cooling Block, Techne, Staffordshire, UK) set at 38 °C for 1 min. A volume of 2 μL of this solution was then placed onto a pre-warmed slide (38 °C) and gently covered with a cover glass (22 × 22 mm). At least 300 sperm were then video-captured in different fields. The subsequent sperm motility traits derived were total motility (MOT, %), progressive motility (PMOT, %), curve–linear velocity (VCL, μm/s), straight-line velocity (VSL, μm/s), average path velocity (VAP, μm/s), amplitude of lateral head displacement (ALH, μm), linearity (LIN, %), straightness (STR, %), wobble (WOB, %), and beat-cross frequency (BCF, Hz).

Parallel to this, mass sperm motility was assessed using the method described by Allen and Champion [14] and Sontakke et al. [28]. Briefly, a drop of 20 μL neat semen was loaded on a glass slide less than 20 min after collection and assessed using a 20× microscope phase-contrast objective at 38 °C. Sperm motility was scored from 1 to 5 as follows: a score of 1 indicated <20% motile sperm distinguished by a very slight to no movement of sperm; a score of 2 indicated between 20 and 40% of sperm showing slow movement but not fast movement; a score of 3 corresponded to 40–60% of sperm showing slow motile whirls, with few immotile sperm visible; a score of 4 indicated 60–80% of sperm showing a formation of rapid moving whirls; a score of 5 corresponded to 80–100% of sperm showing the formation of extremely rapid moving whirls. For consistency, scoring was performed by the same operator.

### 2.4. Statistical Analysis

The data were analyzed using SAS version 9.3 [29]. A point biserial correlation was performed to evaluate the correlation between mass sperm motility and CASA traits. Generalized Linear Mixed Models (GLMM) were used to evaluate the differences in mass sperm motility scores, semen characteristics (semen volume, sperm concentration, and sperm output), and CASA traits between the months of semen collection. Sperm concentration was entered as a linear covariate in the analysis for semen characteristics as well as CASA traits, while male age (young males: ≤4 years; older males: ≥5 years) was entered as a fixed effect. For mass sperm motility, a similar GLMM was performed with a multinomial distribution and a logit link function. Semen variables that were not normally distributed were transformed using the square root transformation, while percentages were arcsine transformed. The significance level was set at *p* < 0.05, and differences between means were compared using Tukey pairwise comparisons.

Repeatability estimates across years (pe^2^—long-term/permanent environmental effects) and within years (te^2^—short-term/temporary environmental effects) for mass sperm motility, CASA traits, and semen characteristics were derived using ASReml software [30]. Statistically, pe² modelled unique animal effects across years. This parameter is also referred to as repeatability, for its obvious application to current-flock selection. In contrast, the te² component separated within-year animal effects from those present across years, thereby ensuring unbiased pe² estimates. For the mass sperm motility scores, data from the present study were combined with those of [18,19], resulting in 1074 records and 28 males being used. For the CASA traits, data from the present study were combined with those of [24], resulting in 480 records and a total of 18 males being used.

## 3. Results

### 3.1. Descriptive Statistics

The mean (± SD) semen volume, sperm concentration, and total number of sperm per semen sample were 1.49 ± 1.04 mL, 2.68 ± 0.70 × 10^9^ sperm/mL, and 4.14 ± 3.03 × 10^9^ sperm cells per sample, respectively. The mean mass sperm motility score was 3.75 ± 1.52. The mean PMOT and MOT recorded were 45.4 ± 14.84% and 78.7 ± 22.90%. Other CASA traits such as VCL, VSL, VAP, ALH, LIN, STR, WOB, and BCF recorded means of, respectively, 75.7 ± 10.7 μm/s, 41.6 ± 5.89 μm/s, 63.9 ± 19.2 μm/s, 2.39 ± 0.39 μm, 55.5 ± 8.26%, 66 ± 10.4%, 84.3 ± 4.21%, and 7.13 ± 1.04 Hz.

### 3.2. Effect of Month and Male Age on Semen Characteristics

There was no significant effect of the month of semen collection on semen volume and total number of spermatozoa per ejaculate (*p* > 0.05; Table 1). However, ejaculates collected in February, July, and August had a higher sperm concentration than ejaculates collected in September, October, and December (*p* < 0.05; Table 1). No significant differences in sperm concentration between ejaculates collected in February, July, and November were recorded (*p* > 0.05; Table 1). Semen characteristics were independent of male age and its interaction with month (*p* > 0.05).

### 3.3. Correlations between CASA Traits and Mass Sperm Motility Scores

Significant positive correlations were recorded between mass sperm motility scores and PMOT, MOT, VCL, and VAP (*p* < 0.0001; Table 2). In contrast, mass sperm motility scores were negatively correlated to LIN, STR, and BCF (*p* < 0.05). Finally, no correlation was recorded between mass sperm motility and VSL, ALH, and WOB (*p* > 0.05).

### 3.4. Repeatability Estimates for CASA Traits, Mass Sperm Motility, and Semen Characteristics

The repeatability estimates for mass sperm motility and CASA traits are indicated in Table 3. The CASA traits had estimates ranging between 0.02 (BCF) and 0.14 (WOB) for long-term/permanent environmental effects (pe^2^), while short-term/temporary environmental effects (te^2^) estimates ranged from 0.01 (PMOT) to 0.10 (BCF). However, the estimates for mass sperm motility were higher (0.17) for pe^2^ than the estimates for te^2^ (0.03). Semen characteristics (semen volume, sperm concentration, and total number of sperm per ejaculate) depended more on pe^2^, ranging from 0.11 (semen volume) to 0.46 (sperm concentration), compared to te^2^ (ranging from 0.09 to 0.11; Table 3).

### 3.5. Effect of the Month on CASA Traits and Mass Sperm Motility

The monthly means for mass sperm motility and CASA traits are presented in Table 4. Mass sperm motility did not differ significantly between the months of collection (*p* > 0.05). Among the CASA traits, PMOT was the highest in semen samples collected in October and September compared to November samples (*p* < 0.05). Furthermore, PMOT was higher in October compared to August (*p* < 0.05). However, semen samples collected in February, July, September, October, and December did not differ in PMOT (*p* > 0.05). MOT was the highest in September when compared to February and November (*p* < 0.05), while it did not differ between July, August, September, October, and December (*p* > 0.05). Other CASA traits were independent of the month of semen collection (*p* > 0.05).

## 4. Discussion

Overall, this study recorded some positive correlations between mass sperm motility scores and CASA traits. The effect of the month of semen collection was reported on some of the quantitative and qualitative attributes of the ejaculate. However, mass sperm motility scoring did not differ between the months of semen collection. Lastly, sperm motility traits recorded by both methods were slightly to moderately repeatable between males across years.

The finding that mass sperm motility scoring was positively correlated to CASA traits is consistent with previous studies on pigeon and cockatiels where sperm motility, as measured by conventional methods, was correlated with CASA traits [31,32]. The positive correlations in the present study indicate that semen samples with higher mass sperm motility scores will also record higher sperm motility when assessed by CASA methods, VCL, and VAP values. Interestingly, some CASA traits such as VSL, VAP, BCF, and PMOT have also been correlated with fertility [7] or egg fertility traits (i.e., sperm quantified in the outer perivitelline membrane and holes made by sperm in the inner perivitelline membrane of an egg) in various strains of Japanese quail [9]. This suggests that, although mass sperm motility scoring is a subjective scoring method, it can be used to assess ostrich sperm motility accurately and to a similar extent as when using CASA technology. Furthermore, CASA traits such as BCF, STR, and VSL were positively correlated to egg fertility traits (i.e., sperm trapped in the outer perivitelline membrane of the egg) but not to fertility after artificial insemination of ostriches [33]. Unexpectedly, BCF, which is the measure of the time needed by the sperm flagellar beat to change its pattern [34], was negatively correlated to mass sperm motility scores. However, this may be due to the fact that no clear pattern of individual sperm flagellar- or head-specific movement could be detected by the mass sperm motility scoring method, suggesting that it may be challenging to predict BCF from mass sperm movement. It thus remains to be tested whether mass sperm motility and fertility are correlated in ostriches, as demonstrated in poultry [14]. It is crucial to establish whether a correlation exists between mass sperm motility and fertility of eggs after artificial insemination. Generally, fertility in naturally mated ostrich flocks has been reported previously, ranging between 77.8 and 87.5% [35,36,37]. On the other hand, fertility in ostriches after artificial insemination was recently reported to reach an average of 38%, ranging between 0 and 82% between females [33]. Offspring production after artificial insemination was studied per female per batch (set of eggs collected and stored not longer than a week before incubation) and averaged 0.06, compared to 1.55 for naturally mated females in a pair breeding system (one male/one female). These results could still be improved as suggested by Muvhali [33]. The male effects on chick survival are not easy to measure, as the eggs are artificially incubated, and the breeding birds do not raise offspring. Therefore, multiple factors such as hatchery management, egg weight, day-old chick weight, and husbandry practices may play a substantial role in offspring survival [2].

Mass sperm motility did not differ significantly between the months of semen collection. This result is consistent with the study of Rybnik et al. [17]. The average score for mass sperm motility in this study (3.75) was, however, lower than those reported by Rybnik et al. [16,17] and Bonato et al. [18,19], which ranged from 4.24 to 4.56. These differences between studies might be explained by the researchers’ variation, subjectivity of the scoring method, year and laboratory variations, as well as by the 30–60% variation in the results from traditional methods of sperm motility assessment previously reported [11,17]. However, since mass sperm motility is commonly measured in undiluted semen, care should be taken to score the semen sample soon after collection to avoid agglutination and subsequent deterioration of semen quality [20]. Among the CASA traits, only PMOT and MOT were affected by the month of semen collection, which is consistent with the report by Smith et al. [21].

This study did not record a monthly effect on semen volume and sperm output. The findings that semen volume and sperm output were not affected by the collection month are inconsistent with previous report [19]. Bonato et al. [19] revealed that the highest semen volume was collected in October and November, as well as in April, while a higher sperm output was evident in September and October. Sperm concentration in the current study was affected by the month of collection and was high in February, July, August, and November. The result that sperm concentration was high in spring–summer months in the Southern Hemisphere concurs with that of Bonato et al. [19] who recorded high sperm concentration around those periods. Furthermore, Muvhali et al. [24] recorded high sperm concentration around the autumn and spring equinox and summer solstice than around the winter solstice in the Southern Hemisphere. The current study recorded high PMOT and MOT in semen collected in September. The latter findings on sperm motility are inconsistent with previous literature where no effect of the collection month was recorded on sperm motility as measured by the CASA system [24]. Although there are moderate discrepancies across studies in semen characteristics, it seems that semen characteristics are a product of a degree of variation throughout months and years which could also be influenced by environmental variation. This is supported by the finding that semen characteristics varied between years of collection [19].

The repeatability estimates derived in this study for CASA sperm traits were low but comparable with the preliminary estimates reported for semen volume and sperm concentration by Cloete et al. [26]. Furthermore, pe^2^ predominated for semen volume, although the derived estimate was lower than the estimate of 0.38 reported by Cloete et al. [26]. No repeatability estimates for either mass sperm motility or CASA traits were, however, available for ostriches to compare with the results of the present study. Finally, most variables recorded in this study were shown to exhibit variation between individuals in previous studies, suggesting that a careful screening of males for superior semen characteristics may be possible [17,18,19,24,25,38]. Phenotypic variation that can be linked to a significant (*p* < 0.05) repeatable portion depicting variation between animals can be used for current-flock selection at least. As repeatability is considered the upper boundary for heritability, it also follows that some traits may exhibit genetic variation. The eventual partitioning of variation between animals in heritable and permanent environmental variances ought to be a priority for further research as more data accrue.

## 5. Conclusions

Mass sperm motility was positively correlated to some sperm motility and sperm velocity traits derived from CASA technology. Resource-limited ostrich farmers may thus utilize the traditional mass sperm motility scoring method to evaluate males for breeding purposes with relative accuracy. Further research is required to derive correlations of traits defined by these methods with the fertility of eggs after artificial insemination and/or natural mating. Finally, mass sperm motility and CASA traits were lowly–moderately repeatable, while semen characteristics were moderately repeatable. These findings suggest that current-flock selection for improved semen characteristics may be possible in male ostriches trained for artificial insemination purposes. Future studies need to partition between-animal variation in its heritable (i.e., genetic) and permanent environmental components to give an indication of gains across generations.

## Figures and Tables

**Table 1 animals-12-01104-t001:** Least-square means (±SEM) of semen volume, sperm concentration, and total sperm output per ejaculate in relation to the month of semen collection from 9 South African Black ostrich males.

Month	Semen Volume (mL)	Sperm Concentration (×10^9^/mL)	Total Number of Sperm (×10^9^)
February	1.27 ± 0.19	2.88 ± 0.14 ^bc^	3.85 ± 0.58
March	1.91 ± 0.35	-	-
July	1.52 ± 0.19	2.91 ± 0.17 ^bc^	4.92 ± 0.64
August	1.51 ± 0.21	3.12 ± 0.17 ^c^	4.78 ± 0.73
September	1.53 ± 0.19	2.21 ± 0.10 ^a^	4.16 ± 0.70
October	1.44 ± 0.19	2.43 ± 0.11 ^a^	3.51 ± 0.45
November	1.53 ± 0.24	2.75 ± 0.09 ^b^	4.61 ± 0.69
December	1.55 ± 0.31	2.34 ± 0.08 ^a^	3.67 ± 0.73
*p*-value	0.952	<0.0001	0.453

^a,b,c^ Least-square means with different superscript within a column differ significantly (*p* < 0.05).

**Table 2 animals-12-01104-t002:** Correlation between mass sperm motility scores and CASA traits (*N* = 180, 9 males). Mass sperm motility was measured on a scale from 1 to 5 (1, <20% motile sperm distinguished by a very slight to no movement of sperm; 2, between 20 and 40% sperm showing slow movement but not fast; 3, 40–60% sperm showing slow motile whirls with few immotile sperm visible; 4, 60–80% sperm showing the formation of rapid moving whirls; 5, 80–100% sperm showing the formation of extremely rapid moving whirls).

CASA Traits	Mass Sperm Motility
ALH (μm)	0.13
BCF (Hz)	−0.20 **
LIN (%)	−0.29 ***
MOT (%)	0.59 ***
PMOT (%)	0.54 ***
STR (%)	−0.33 ***
VAP (μm/s)	0.34 ***
VCL (μm/s)	0.34 ***
VSL (μm/s)	0.04
WOB (%)	0.15

*** *p* < 0.0001, ** *p* < 0.05. ALH, amplitude of lateral head displacement; BCF, beat-cross frequency; LIN, linearity; MOT, total motility; PMOT, progressive motility; STR, straightness; VAP, average path velocity; VCL, curve-linear velocity; VSL, straight-line velocity; WOB, wobble.

**Table 3 animals-12-01104-t003:** Repeatability estimates (± SE) for mass sperm motility, CASA traits, and semen characteristics (semen volume, sperm concentration, and total number of sperm per ejaculate). CASA traits and semen characteristics *N* = 480 and mass sperm motility scores: *N* = 1074. These results were derived from ejaculates collected from up to 28 male ostriches from a combination of records collected for the present study and those of Bonato et al. [18,19] as well as Muvhali et al. [24].

Variables	pe^2^	te^2^
Mass sperm motility	0.17 ± 0.06	0.03 ± 0.02
ALH (μm)	0.11 ± 0.05	0.07 ± 0.04
BCF (Hz)	0.02 ± 0.03	0.10 ± 0.04
LIN (%)	0.04 ± 0.03	0.01 ± 0.03
MOT (%)	0.04 ± 0.03	-
PMOT (%)	0.09 ± 0.05	0.01 ± 0.03
STR (%)	0.07 ± 0.04	0.05 ± 0.04
VAP (μm/s)	0.07 ± 0.04	0.07 ± 0.04
VCL (μm/s)	0.06 ± 0.04	0.08 ± 0.05
VSL (μm/s)	0.06 ± 0.04	0.05 ± 0.04
WOB (%)	0.14 ± 0.06	0.06 ± 0.04
Semen volume (mL)	0.11 ± 0.06	0.09 ± 0.04
Sperm concentration (×10^9^/mL)	0.46 ± 0.10	0.09 ± 0.04
Total number of sperm (×10^9^)	0.37 ± 0.10	0.11 ± 0.04

pe^2^—long-term/permanent environmental effects, te^2^—short-term/temporary environmental effects. ALH, amplitude of lateral head displacement; BCF, beat-cross frequency; LIN, linearity; MOT, total motility; PMOT, progressive motility; STR, straightness; VAP, average path velocity; VCL, curve-linear velocity; VSL, straight-line velocity; WOB, wobble.

**Table 4 animals-12-01104-t004:** Least-square means (± SEM) of mass sperm motility and CASA traits for the duration of the study period in semen collected from 9 South African Black ostrich males.

Sperm Motility Traits	February	March	July	August	September	October	November	December	*p*-Value
Mass motility	3.54 ± 0.31	4.40 ± 0.43	3.40 ± 0.29	3.91 ± 0.29	3.90 ± 0.26	4.13 ± 0.30	3.36 ± 0.38	3.90 ± 0.33	0.234
ALH (μm)	2.49 ± 0.10	-	2.44 ± 0.10	2.25 ± 0.11	2.37 ± 0.11	2.51 ± 0.11	2.27 ± 0.11	2.49 ± 0.13	0.566
BCF (Hz)	7.32 ± 0.26	-	7.18 ± 0.25	7.41 ± 0.29	7.08 ± 0.29	6.63 ± 0.28	7.25 ± 0.27	7.11 ± 0.35	0.858
LIN (%)	56.8 ± 1.57	-	52.5 ± 1.58	57.6 ± 1.83	52.5 ± 1.84	45.5 ± 1.73	61.5 ± 1.67	56.8 ± 2.26	0.199
MOT (%)	69.7 ± 4.35 ^ab^	-	85.7 ± 4.41 ^bc^	78.9 ± 5.10 ^abc^	90.1 ± 5.14 ^c^	87.9 ± 4.82 ^bc^	61.0 ± 4.63 ^a^	72.5 ± 6.35 ^abc^	0.0004
PMOT (%)	41.1 ± 3.32 ^abc^	-	45.9 ± 3.28 ^abc^	39.5 ± 3.34 ^ab^	51.7 ± 3.70 ^bc^	52.1 ± 3.58 ^c^	36.7 ± 3.49 ^a^	43.3 ± 4.45 ^abc^	0.001
STR (%)	68.9 ± 2.11	-	63.6 ± 2.10	69.1 ± 2.39	61.1 ± 2.38	53.3 ± 2.29	72.8 ± 2.22	66.3 ± 2.87	0.171
VAP (μm/s)	60.4 ± 2.16	-	62.8 ± 2.15	55.2 ± 2.46	68.5 ± 2.45	74.0 ± 2.35	61.6 ± 2.27	69.1 ± 2.98	0.110
VCL (μm/s)	72.4 ± 2.15	-	75.7 ± 2.16	66.1 ± 2.48	79.7 ± 2.49	86.4 ± 2.36	72.7 ± 2.28	80.8 ± 3.05	0.076
VSL (μm/s)	40.6 ± 1.14	-	39.7 ± 1.17	38.3 ± 1.35	41.4 ± 1.37	38.7 ± 1.27	44.7 ± 1.22	45.7 ± 1.69	0.838
WOB (%)	82.4 ± 1.35	-	82.6 ± 1.33	82.9 ± 1.43	85.6 ± 1.41	84.9 ± 1.40	84.1 ± 1.38	85.2 ± 1.58	0.367

^a,b,c^ Least-square means with different superscripts within a row differ significantly (*p* < 0.05). ALH, amplitude of lateral head displacement; BCF, beat-cross frequency; LIN, linearity; MOT, total motility; PMOT, progressive motility; STR, straightness; VAP, average path velocity; VCL, curve-linear velocity; VSL, straight-line velocity; WOB, wobble.

## Data Availability

The data supporting the results reported in this study is owned by the Western Cape Department of Agriculture and could be available upon request.

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
