# Peer review of "Mass Sperm Motility Is Correlated to Sperm Motility as Measured by Computer-Aided Sperm Analysis (CASA) Technology in Farmed Ostriches"

_animals, 2022, doi:10.3390/ani12091104_

Round 1
Reviewer 1 Report
This paper describes the comparative semen evaluation between human/microscope assessment and evaluation by CASA for ostrich males.
The point is well made that while the CASA analysis may be 'ideal' and non-subjective, it is also not always practical, especially for farmed species. An inclusion of more information on the business (numbers held, numbers produced, the economics of the species for food consumption, etc) of ostrich farming - both in South Africa and globally - would be welcome in the introduction to highlight the importance of understanding the reproductive biology of this species.
There are a lot of data in table 4, making it difficult to read. Suggest spacing out the lines more and/or consider reducing significant figures to 1.
A good deal of the info in the first paragraph of the discussion is results. A recap of the highlights here is fine, but the details of the results should be kept in the results section.
Lines 253-254. What is known about fertility in ostriches? What is normal/average offspring production? Are there any known male effects on offspring survival or thriftiness?
Author Response
Reviewer 1:
This paper describes the comparative semen evaluation between human/microscope assessment and evaluation by CASA for ostrich males.
The point is well made that while the CASA analysis may be 'ideal' and non-subjective, it is also not always practical, especially for farmed species. An inclusion of more information on the business (numbers held, numbers produced, the economics of the species for food consumption, etc) of ostrich farming - both in South Africa and globally - would be welcome in the introduction to highlight the importance of understanding the reproductive biology of this species.
We thank you for the positive views towards our study and to ostrich farming in general. Literature on the background of ostrich farming in South African, economy, food contribution and global contribution has been added (Line 43-48).
There are a lot of data in table 4, making it difficult to read. Suggest spacing out the lines more and/or consider reducing significant figures to 1.
The data in Table 4 has been presented in a landscape format to improve readability (Line 247-249). Means above 10 are also presented with only one decimal.
A good deal of the info in the first paragraph of the discussion is results. A recap of the highlights here is fine, but the details of the results should be kept in the results section.
The paragraph has been revised and was presented to capture the highlights of the results (Line 254-259).
Lines 253-254. What is known about fertility in ostriches? What is normal/average offspring production? Are there any known male effects on offspring survival or thriftiness?
Generally, fertility in naturally mated ostrich flocks has been reported previously ranging between 77.8 to 87.5 % (Deeming, 1995; Van Schalkwyk et al., 2000; Bunter et al., 2001). On the other hand, fertility in ostriches after artificial insemination was reported recently to reach an average of 38%, ranging between 0-82% between females (Muvhali, 2022). Offspring production after artificial insemination was studied per female per batch (set of eggs collected and stored not longer than a week before incubation) and averaged 0.06, compared to 1.55 for naturally mated females in a pair breeding system (1 male: 1 female). These results could still be improved as suggested by Muvhali (2022).Male effects on chick survival are not easy to measure as the eggs are artificially incubated and the breeding birds do not raise offspring. Therefore, multiple factors such as hatchery management, egg weight, day old chick weight and husbandry practices may play a huge role on offspring survival. This information has been added in the manuscript (Line 281-291).
References
Muvhali, P.T. 2022. The refinement of an artificial insemination protocol in ostriches. PhD dissertation. Stellenbosch University, South Africa.
Deeming, D.C. Factors affecting hatchability during commercial incubation of Ostrich (Struthio camelus) eggs. Br. Poult. Sci. 1995, 36, 51-65.
Van Schalkwyk, S.J.; Cloete, S.W.P.; Brown, C.R.; Brand, Z. Hatching success of ostrich eggs in relation to setting, turning and angle of rotation. Br. Poult. Sci. 2000, 41, 46-52.
Bunter, K.L.; Cloete, S.W.P.; Van Schalkwyk, S.J.; Graser, H.-U. Factors affecting reproductive performance in farmed ostriches. Proc. Assoc. Advmt. Anim. Breed. Genet. 2001, 14, 43-46.
Reviewer 2 Report
This scientific manuscript adds a positive contribution for semen evaluation in Farmed Ostriches. However, description of results is somewhat confuse. We see results in tables that don´t match with it descriptions in text. Only when there are different superscripts, results are different (P<0.05). We see in 1 and 4, common superscripts in some results, but in text it is described as different. Many confusions were detected. In table 1, we see in the botton of the table, different superscripts with a row??; (line 172); perhaps in the column?; In line 176, we see negative correlations between mass sperm motility and BCF ; is it expected?; . in table, 3, lines 201-202, Can you detail and explain the N Values, N=480 and N=1074; English text (legend) (lines 214-215) is confuse?; Lines 210-212, are not entirely correct, based on superscripts letters; In MOT and PMOT (table 4), differences among months (see common letters) don´t match with results description. This section is not correct and confuse; Also the same motif, see, lines 211-212, 226, 228, 255, 277, 270, 291 305.; lines 265-266 is perhaps not entirely correct?; Explain why results with low heritability (h2) and repeatability may me "good" for selection?; i don´t understand; Is it possible to explain in text the difference among pe2 vs te2 (lines 203-204); Which is more important? ; see punctuation error, line 270;
Author Response
Reviewer 2:
This scientific manuscript adds a positive contribution for semen evaluation in Farmed Ostriches. However, description of results is somewhat confuse. We see results in tables that don´t match with it descriptions in text. Only when there are different superscripts, results are different (P<0.05). We see in 1 and 4, common superscripts in some results, but in text it is described as different. Many confusions were detected. In table 1, we see in the bottom of the table, different superscripts with a row??; (line 172); perhaps in the column?
We thank the reviewer for this. The mistake has been rectified in the manuscript and row has been changed to column (Line 200). The interpretation of the results has been rectified in Table 1 (Line 191-195) and Table 4 (Line 239-244).
In line 176, we see negative correlations between mass sperm motility and BCF ; is it expected?
The negative correlation between mass sperm motility and BCF was not expected. However, this may be due the fact that no clear pattern of individual sperm flagellar or head specifically could be detected by the mass sperm motility scoring method, suggesting that the prediction of this trait from mass sperm movement may be challenging. Text to this effect has been added in the manuscript (Line 273-278)
In table, 3, lines 201-202, Can you detail and explain the N Values, N=480 and N=1074
These values have been explained in line 176-179. These were data generated from previous studies that our group conducted on the same resource flock.
English text (legend) (lines 214-215) is confuse?
The English has been revised in the text.
Lines 210-212, are not entirely correct, based on superscripts letters. In MOT and PMOT (table 4), differences among months (see common letters) don´t match with results description. This section is not correct and confuse
Also the same motif, see, lines 211-212, 226, 228, 255, 277, 270, 291 305.
This has been revised in the manuscript (line 239 – 244)
Lines 265-266 is perhaps not entirely correct?
This has been revised to avoid confusion (Line 302)
Explain why results with low heritability (h2) and repeatability may me "good" for selection?
If a trait is variable and repeatable, it is expected to respond to current-flock selection. This means that current-flock gains can be made if the best performers are retained in the flock. The partitioning of genetic and animal PE variances will become feasible as more data are collected, informing analysts on selection gains in future generations. These thoughts were added in the text (Line 330-335)
I don´t understand; Is it possible to explain in text the difference among pe2 vs te2 (lines 203-204); Which is more important? ; see punctuation error, line 270;
The difference between the two was explained in line (173-176) under the statistical analysis section. Both are important as they indicate whether the trait is affected by short-term environmental effects or by the long-term effect. Both parameters should be included in the model of analysis, as informed by log-likelihood ratios. Temporary environmental effects do not have value in current-flock selection, but need to be modelled. If not considered pe effects will be biased upwards, resulting in inflated estimates for current-flock gains.
The punctuation error has been rectified (Line 307)
Reviewer 3 Report
CASA technology and mass sperm motility scoring are the two main methods currently used in a wide range of livestock species for the assessment of sperm motility. Results of CASA are more accurate than mass sperm motility. However, mass sperm motility method is cheap. This manuscript demonstrated that mass sperm motility is correlated to sperm motility as measured by computer aided sperm analysis technology in Farmed Ostriches. This finding is useful for the evaluating semen quality. There are two major concerns for this manuscript. 1. The conclusion part is not concise; many of them belong to discussion part. Please try to summarize the main finding. 2. For abstract and introduction part, the author mainly introduced the relationship between the two methods and relative results, but did not introduce other content. For example, results in figure 1, figure 3 and figure 4 were not introduced in the abstract, indicating the important data information of this study is not too much.Author Response
Reviewer 3:
CASA technology and mass sperm motility scoring are the two main methods currently used in a wide range of livestock species for the assessment of sperm motility. Results of CASA are more accurate than mass sperm motility. However, mass sperm motility method is cheap. This manuscript demonstrated that mass sperm motility is correlated to sperm motility as measured by computer aided sperm analysis technology in Farmed Ostriches. This finding is useful for the evaluating semen quality. There are two major concerns for this manuscript. 1. The conclusion part is not concise; many of them belong to discussion part. Please try to summarize the main finding.
The conclusion has been shortened as suggested by the reviewer (Line 337-347).
- For abstract and introduction part, the author mainly introduced the relationship between the two methods and relative results, but did not introduce other content. For example, results in figure 1, figure 3 and figure 4 were not introduced in the abstract, indicating the important data information of this study is not too much.
The information relevant to results reported in table 1, 3 and 4 has been included in the introduction (Line 87-100). In the abstract, we were compelled to only mention the key results for this study as we are limited by the word count requirement.
Round 2
Reviewer 2 Report
Some explanations in discussion could be presented about seminal variations among seasons and months. I think that discussion could be improved with the justifications of some results and correlations